# Dietary Cinnamon Bark Affects Growth Performance, Carcass Characteristics, and Breast Meat Quality in Broiler Infected with *Eimeria tenella* Oocysts

**DOI:** 10.3390/ani12020166

**Published:** 2022-01-11

**Authors:** Mohammed M. Qaid, Saud I. Al-Mufarrej, Mahmoud M. Azzam, Maged A. Al-Garadi, Abdulmohsen H. Alqhtani, Abdulaziz A. Al-abdullatif, Elsayed O. Hussein, Gamaleldin M. Suliman

**Affiliations:** Animal Production Department, College of Food and Agriculture Sciences, King Saud University, Riyadh 11451, Saudi Arabia; salmufarrej@ksu.edu.sa (S.I.A.-M.); mazzam@ksu.edu.sa (M.M.A.); malgaradi@ksu.edu.sa (M.A.A.-G.); aalabdullatif@ksu.edu.sa (A.A.A.-a.); shessin@ksu.edu.sa (E.O.H.); gsuliman@ksu.edu.sa (G.M.S.)

**Keywords:** breast, broiler, carcass traits, cinnamon bark, meat quality

## Abstract

**Simple Summary:**

Antimicrobial tolerance problems have culminated in an increased focus on raising broiler chickens without using any antibiotics, and an increasing interest has developed in non-antibiotic feed supplements with potential productivity and health benefits. Previous studies have shown beneficial results linked to the addition of cinnamon to broiler diets under health conditions without induced coccidiosis. In this study, different amounts of dietary cinnamon, a phytogenic of the Lauraceae family, were evaluated for their effects on carcass characteristics and meat quality in broilers infected with oocysts of *Eimeria tenella*. Overall, emeriosis negatively affects bird slaughter weight, carcass yield, and most carcass traits. Compared with the positive control, cinnamon increased slaughter weight, carcass yield, and the percentage weights of heart, proventriculus, gizzard, breast, and pancreas. In addition, cinnamon at 2 g/kg diet improved performance, cocking loss, and meat tenderness among cinnamon groups. The ionophore salinomycin (Sacox^®^) group had the highest slaughter yield, myofibril fragmentation index (MFI), and texture profile analysis (TPA) of meat. The current research offers equivalent and unbiased findings from a study of substitutes for commercial coccidiostats in a consistent experimental paradigm that applies well to commercial conditions.

**Abstract:**

A total of 150 broiler chicks were used to determine the impact of dietary cinnamon bark powder (CBP; *Cinnamomum verum*) on breast meat quality, growth performance, and carcass characteristics of birds under coccidiosis, as one of the protozoan parasitic diseases. A total of 5 replicates of birds received 1 of the following 6 groups for 34 days: control groups (1 and 2) received a basal diet without the addition of CBP or salinomycin; group 3 received a basal diet with 0.066 g salinomycin; groups 4–6 were given a basal diet supplemented with 2, 4, and 6 g CBP/kg feed, respectively. On day 21, 4 × 10^4^/100 µL of *Eimeria tenella* oocysts/bird were challenged, except for the negative control (NC). At the end of the experimental trial, five birds/group were sampled for carcass characteristics and breast attributes. Overall, emeriosis negatively affects slaughter body mass, carcass yield, and the majority of carcass characteristics in birds, and cinnamon can mitigate these effects. Cinnamon groups, particularly at the 2 g level, alleviated the negative effect on performance caused by coccidia infestation to the same or greater extent as the negative control and salinomycin treatment groups. Furthermore, when compared with the other experimental groups, the addition of cinnamon improved some physicochemical properties with some affecting meat quality, such as decreasing MFI and increasing toughness in cinnamon-treated groups. In summary, it can be concluded that CBP can enhance the shelf life, carcass, and quality of birds’ meat by maximizing the productive performance efficiency and breast meat productivity of birds under coccidiosis infestation. Further research is required to investigate the use of cinnamon to optimize the quality of meat and the productivity of both healthy and diseased broilers.

## 1. Introduction

Total imports of Saudi Arabian chicken meat were reduced from 618,000 to 520,000 tons in 2020 and 2021 as local production of Saudi Arabian chicken meat increased. Globally, increases in broiler meat production were also expected, with ~100.827 million tons produced in 2020 and projections for 2022 of ~100.9 million tons [1]. These increases in production are related to the high demand for poultry meat, which exceeds that of other animals, reflecting its rich nutritional value and favorable profit margin. As a result of the sharp increase in the production and popularity of broiler meat, the demand for improving the palatability and quality of meat has increased [2]. Important criteria for assessing the slaughter value of poultry carcasses include the percentage of dressing, the percentage of culinary, the tenderness of the meat, and the muscle content of the carcass. According to [3], broilers have a greater dressing percentage, reaching ~75%. This indicator is considered to be essential for evaluating the quantity of produce. Likewise, the composition of the carcass is essential because a high muscle content, particularly of the breast muscle, and a low fat content stimulate the consumer’s appetite. While the quality of the meat comprises a sequence of attributes, the consumer is most concerned about the tenderness, color, juiciness, and palatability of the meat [4,5,6]. Meat color is a simple and quick criterion for evaluation and seems to be an important indicator to consumers. Fresh chicken meat should be light red in color. Thus, at first glance by a consumer, any change in color indicates declining nutritional value and deterioration in quality. The capacity of bird meat to retain its natural and added moisture during processing, storage, and when subjected to an external force is termed its water-holding capacity (WHC). Measuring WHC is one of the easiest ways to detect the quality of meat, and it has a direct impact on yield and appearance [7].

The larger body and breast weights of birds are related to larger areas and diameters of muscle fibers (myofiber), and older birds have lower myofiber density than younger birds [8]. Consumers expect tenderness; therefore, measuring tenderness is an appropriate way of measuring consumer satisfaction in eating meat products, and it is subjectively judged to be a sense of the hardness or elasticity of tough or tender meat [9]. Texture profile analysis (TPA) is a constructive technique that mimics the bite action of the mouth through a popular double compression test to determine the textural properties of foods, and it does not require a large number of investigators to assess the texture of poultry meat [10].

Humans that consume poultry meat could be directly affected by antibiotic growth promoters (AGPs) residues in the meat, or indirectly affected by the development of antibiotic-resistant pathogens in the meat [11]. Some researchers have investigated the growth and meat quality of birds fed diets containing natural herbs or extracts of natural herbs [12,13,14,15,16]. Coccidiosis is a parasitic disease caused by a protozoan that causes enteritis, hemorrhagic cecal lesions, and bloody diarrhea, with significant economic losses worldwide to the poultry industry [17].

Cinnamon spice is obtained from the inner bark of *Cinnamomum verum*, which is a vigorous, evergreen, annual, and aromatic ethnomedicinal plant belonging to the Lauraceae family. Various herb extracts, including cinnamon plant oils and their bioactive constituents, such as cinnamaldehyde and eugenol, are used as dietary supplements in poultry production. These extracts have a variety of uses, including antibacterial activity against many pathogens and acceleration of the growth of good bacteria, such as lactic acid bacteria and bifidobacterial, in the poultry intestine [18,19]. Furthermore, cinnamon oil has potent hypercholesterolemia, anticandidal, antioxidant, analgesic, and antiulcer activities [20], and the health-promoting and performance-enhancing effects of dietary aromatic herbs and extracts have been shown in both healthy and diseased farm animals [21]. Dietary cinnamon supplementation improves the quality and shelf life of broiler meat and maximizes meat productivity by lowering abdominal fat, increasing the dressing percentage and redness, and reducing drip loss and antioxidant activity of the breast meat of stressed broilers [22]. The leaves and bark of the cinnamon herb and its metabolites are receiving more attention as phytogenic feed additive substitutes for AGPs because they are natural antibiotics—readily available, non-toxic, and residue-free [20]—as well as enhancing poultry growth and improving carcass characteristics and the quality of broiler meat, and enhancing immunity and microbiological factors. However, limited studies are available on the use of cinnamon powder as a phytogenic alternative antibiotic and potential nutrition enhancer in the diets of broilers facing coccidiosis challenges.

This study hypothesized that the cinnamon herb could be used to enhance growth performance, carcass traits, meat quality, and physico-chemical properties of the breast muscle of broiler chickens experiencing coccidiosis infestation. Various levels of cinnamon bark plant were evaluated to validate this hypothesis. Therefore, this study aimed to test the ability of the cinnamon powder to be used as a dietary AGP substitute for enhancing the breast quality, carcass characteristics, and growth performance of Ross broiler chickens challenged by *Eimeria tenella*.

## 2. Materials and Methods

### 2.1. Ethical Approval

The research was carried out in accordance with the Kingdom of Saudi Arabia’s ethical standards for animal use (Ethic committee King Saud University, Ethical approval number: KSU-SE-20-44).

### 2.2. Infection with Eimeria tenella

Our previous study [23] described the source of *Eimeria tenella* (*E. tenella*) oocysts, the sporulation of unsporulated oocysts, the identification, passage, and propagation of sporulated oocysts, and the inoculation procedure. All birds, with the exception of NC, were administered 1 mL of double distilled water containing 4 × 10^4^/100 µL/bird of live sporulated oocysts of *E. tenella* at 21 days of age, according to previous research [24,25,26]. The authors choose the 7 dpi for anticoccidial indices (data published in [23]). Thus, this study is an extension of the same broiler batch experiment in which the anticoccidial indicators of CBP evaluation, namely the number of fecal oocysts, survival rate, bloody diarrhea, and lesion scores, were included. As a result, this investigation does not address the assessment of anticoccidial indicators. We found that CBP was effective on *E. tenella.* Furthermore, salinomycin, commercially called (Sacox^®^; Huvepharma NV, Belgium), is a standard product that protects birds from coccidiosis. For growth performance sampling, we chose 7 and 14 dpi. Following [27], meat quality and carcass variables were measured in all treatments on the last day of the trial, at 34 days of age (14 dpi here), in order to mimic the carcass traits of commercial poultry at marketing weights and to identify meat quality.

### 2.3. Birds and Husbandry

The study was carried out in a controlled environment—a heated battery room at the Animal Production Department’s experimental poultry research unit at King Saud University, Riyadh, Saudi Arabia. The trial operated during spring month “March–April 2019” with average temperature varying between 20.4 °C and 33.4 °C and the average relative humidity in Riyadh in April was 28% during the experimental period. A total of 150, 1-day-old, mixed-sex, commercial Ross 308 broiler birds were collected from the national commercial hatchery (Al Wadi Poultry Company “Al Khomasia,” Riyadh, Saudi Arabia). At the hatchery, the birds were immunized against Newcastle and infectious bronchitis diseases. The chicks were randomly divided into 30 experimental cages with 5 replicates of 6 treatments with 5 chicks per replicate. At 1 day of age, the temperature was set at 35 °C and gradually decreased by 1 °C every 2 days until a permanent temperature of 22 °C was reached. Then, it was maintained until the end of the trial. Relative humidity ranged from 65–85%. Feed and water were supplied *ad libitum*, and birds were kept on a “23 h on and 1 h off” light schedule.

### 2.4. Experimental Dietary Treatments

Poultry diets were obtained from the Arabian Agricultural Services Company. The ingredients and chemical properties of the commercial starter (1–21 days) and finisher (22–34 days) broiler chicks’ diets were analyzed, formulated, and mixed in a mashed form, based on Ross 308′s recommendation guidelines (Appendix A). On arrival, the birds were randomly allocated to one of the following 6 treatments:

T1—negative control group given an unmedicated diet, these were unchallenged coccidiosis chicks (NC); T2—positive control group given an unmedicated diet + coccidial challenge (PC); T3—medicated diet with salinomycin sodium (66 mg salinomycin/kg diet) + coccidial challenge; T4–T6—2, 4, and 6 g CBP/kg diet, respectively, + coccidial challenge. The supplemented levels of purchased CBP or salinomycin powder were mixed with the basal broiler diet.

### 2.5. Preparation and Compositions: Cinnamon Bark Powder 

*Cinnamomum verum* bark was acquired from a nearby store in Riyadh, Saudi Arabia, for use in this study. The dried bark used was ground to a fine powder. Moreover, as described by [23], high-performance liquid chromatography (HPLC) and gas chromatography–mass spectrometry (GC–MS) were used to detect the biologically relevant compounds in the CBP extract mixture. A total of 26 different active compounds with the highest quality were detected by GC–MS in the CNB extract, particularly Cinnamaldehyde, 3-phenyl-, hexadecanoic acid, (E)-2-propenal, methyl ester, 14-methyl-,methyl ester, pentadecanoic acid, oxime-, methoxy-phenyl-, and 2-methyl-benzofuran, as previously accounted for in [28].

### 2.6. Performance Measurements and Production Efficiency

Body weight and feed intake (FI) of birds were recorded at 1, 7, and 14 dpi per replicate to evaluate performance. Then, for each pen, the body weight gain (BWG) and FI were recorded. The average feed conversion efficiency (FCR) was calculated by dividing FI (g) by BWG (g). The European production efficiency factor (PEF) was calculated as follows:PEF = ((Live weight (kg) × Livability)/(Age in days × FCR)) × 100 (1)

### 2.7. Carcass Relative Weights

A total of 30 34-day-old birds (Ross 308) of each treatment (*n* = 5, 1 male bird per cage, per treatment) were randomly selected for slaughter and subjected to feed withdrawal for 10 h. Before slaughtering, the birds were weighed. After bleeding, birds were defeathered and eviscerated (about two minutes after jugular vein incision). The feathers of the birds were plucked, and the viscera was eviscerated. The carcasses were dissected after the head, feathers, and shanks were removed. Live weight at slaughter (SW, kg) and carcass weight (CW, kg) were registered to calculate the yield of (CY%) (dressing percentage) = (CW/SW) × 100). Then, the CW were dissected into commercial parts, including the offal (heart, liver, and gizzard), neck, abdominal fat, heart, liver, proventriculus, gizzard, lymphoid organs, breast, leg, and pancreas, and weighed. Then, the relative weights of these weights were expressed in relation to the SW.

### 2.8. Meat Quality Indices

In addition to carcass parameters, the left and right parts of the breast meat (pectoralis major) of each selected bird (one breast per cage/treatment) were used for qualitative analyses. For the pH and color measurements, breast samples were kept at 4 °C; meanwhile, for the other quality measurements, they were frozen at −20 °C until further measurements were conducted for other meat quality assessments, and then thawed overnight in the fridge at 4 °C before analysis. The initial and ultimate pH and color components of dissected breast muscle were determined at 1 and 24 h. 

#### 2.8.1. Breast Meat Physicochemical Characteristics (pH, Temperature, and Color Indicators)

The breast pH was measured after processing with a microprocessor Hanna Instruments pH meter, and incisions were made in the cranial left side of the pectoral muscle. At 1 and 24 h, an average of 3 pH measurements for each sample had been taken.

A thermocouple thermometer probe from Eutech Instruments was placed deep in the center of the muscle 1 h postmortem to monitor internal core temperature values of the pectoral muscle. 

Breast flesh color measurements, developed by Dr. Richard Hunter as the Hunter values—lightness (L*), redness (a*), and yellowness (b*)—were set on CIELAB scales and assessed with a Chroma meter, 1 and 24 h after the slaughter on 2 different areas of the inner side of the cranial position of the breast muscles. Values for L*, a*, and b* were converted to estimate the total color change (∆E), Chroma meter (saturation index), hue angle, browning index (BI), and whiteness index (WI), as described by [29,30,31]. According to [32], these measurements obtain a much more accurate assessment of how consumers perceive the color of meat. The averages of the two readings of the color components were taken. 

#### 2.8.2. Water-Holding Capacity

The breast meat water-holding capacity (WHC) of the frozen meat samples was measured immediately after thawing overnight at 4 °C using the compression method outlined by [33].

#### 2.8.3. Cooking Loss 

The cooking loss (CL) was measured as follows [33]: CL was determined by weighing muscle samples with a semi-analytical balance, placing them in a commercial indoor tabletop grill, cooking them until they reached an internal temperature of 75 °C, and reweighing them after cooking. The CL percentage = [(Initial weight − Cooked weight)/Initial weight] × 100.

#### 2.8.4. Drip Loss 

The parts of the breasts used for the drop test were weighed separately, packed, and kept at 4 ± 1 °C for 24 h. Then, the difference in weights before (Wi) and after (Wu) storage was calculated and expressed as a drop loss proportion [34]. Drip loss (DL) (%) = [(Wi − Wu)/Wi] × 100.

#### 2.8.5. Myofibril Fragmentation Index

The procedures mentioned by [33] were used to evaluate myofibril fragmentation (MFI) as an indirect measure of calpain intracellular proteases. An amount of 4 g of muscle, minced with scissors, was homogenized in a blender for 30 s with 40 mL of cold MFI buffer at 2 °C. After many washes, the suspension aliquots were diluted in MFI buffer to a final concentration of 0.5 mg/mL and poured into a cuvette for immediate measurement of absorbance at 540 nm using a spectrophotometer. Each sample’s MFI was calculated to be A_540 nm_ × 200.

#### 2.8.6. Meat Texture Analysis

The shear force (SF) and TPA of the samples were determined using a texture analyzer (TA.HD. Stable Micro Systems, Surrey, UK) in 2 parts per replicate (1 breast/replicate/treatment). After the cooked samples had cooled at 22 °C, 5 round core meat slices (1.27 and 2.5 cm diameter for SF and TPA, respectively) were cut from each sample, parallel to the longitudinal direction of the muscle fibers, using a handheld coring tool. During the SF test, the maximum force (kgf) was applied vertically to the fibers using a TA.HD. Texture Analyzer, designed for a Warner-Bratzler shear blade, with a triple-slotted cutting edge. The speed of the crosshead was set at 200 mm/min. The SF values were estimated from the maximum point of the generated curve. A cylindrical piston (75 mm diameter) was used to compress the TPA sample to within 80% of its original height over two test cycles in 5 s. The texture analyzers conditions were used to generate force–time curves of deformation. The hardness, springiness, chewiness, and cohesion parameters were measured following [35].

### 2.9. Statistical Analysis

In the statistical analysis system [36], a general linear model (GLM) was used to analyze slaughter characteristics and meat quality data. Six groups were arranged in five replicates in a completely randomized design. Each replicate cage represented an experimental unit. On the 34th day of age, male birds were sampled (*n* = 5 birds per treatment, 1 bird from each replicate).

All data were analyzed using one-way ANOVA and expressed as a statistical mean ± standard error of the mean (SE) using the following models:Yij = μ + Ti + Ԑij
where Yij is the observed j parameters in the i^th^ treatment, μ is the overall mean of the measurements, Ti is the effect of the i^th^ treatment, and Ԑij is the random residual error. To assess significant differences between means for measurements using Duncan’s multiple range test, a statistical significance level of *p* < 0.05 was used.

## 3. Results

According to our previous research [23], clinical coccidiosis symptoms were observed in birds after infection with oocysts of *E. tenella*. It was found that 6 g of CBP had moderate anti-coccidial activity and could be used to treat poultry emeriosis in the field. Consequently, CBP decreased the severity of lesions and reduced oocyst excretion per gram in chickens’ droppings. Moreover, based on the HPLC and GC–MS results, we found that cinnamaldehyde and other important bioactive compounds are present in the cinnamon bark extract.

### 3.1. Performance Measurements and Production Efficiency

Table 1 shows that the challenge of coccidiosis had adverse effects on BWG, FCR, and PEF of the birds at 1st and 2nd week post-infection and suffered from it over the entire period compared with an unchallenged control group. During the second week after infection, the BWG, FCR, and PEF of the birds were dose-dependent, increasing as the cinnamon level decreased. Birds receiving a 2g CBP/kg diet had higher BWG, FCR, and PEF over the entire post-infection period (0–14 dpi) than those which received 4g CBP, 6g CBP, and PC (*p* < 0.05), but was similar to those receiving 66 mg salinomycin and NC. This means that birds given 2g CBP/kg gained more and converted feed more efficiently. The statistical models of FI and FCR did not differ significantly (*p* > 0.05) during the first or second weeks or during the entire period after the *Eimeria tenella* oocyst challenge. 

### 3.2. Carcass Characteristic Variables

The effects of the CBP on carcass variables at 34 d of age (14 days post-inoculation) are shown in Table 2. Except for liver, leg, fat, and lymphoid organs (bursa, thymus, and spleen) values, there were significant differences in slaughter variables between treatments. The live weight, CW, and the carcass yield of slaughtered birds were statistically different (*p* < 0.05); this result indicates that the adverse effects of *Eimeria tenella* infection in birds were clearly observed in the PC group and were compensated for in the salinomycin and cinnamon groups. Thus, the heart, proventriculus, gizzard, breast, and pancreas % CW values did differ (*p* < 0.05) between treatments. Birds receiving cinnamon at levels of 4 g with their basal diet had higher percentage heart, proventriculus, gizzard, and pancreas weights, respectively, compared with those in other dietary groups. The lowest abdominal fat yield exhibited an insignificant decrease (*p* < 0.05) with increased CBP compared with the NC group.

### 3.3. Breast Meat Physicochemical Characteristics

Table 3 and Table 4 display the influence of CBP on the physicochemical characteristics of broiler breast samples at 1 and 24 h postmortem, respectively. The core temperature, initial and ultimate lightness, total color change, and WI of samples were different (*p* < 0.05). Otherwise, the initial and ultimate values for pH, redness, yellowness, total color change (Delta E: ∆E), hue angle, BI, and saturation index (Chroma) did not differ among experimental groups (*p* > 0.05). The core temperature values of breast muscle were significantly different (*p* < 0.05), with the broilers from the 2 g of CBP/kg treatment having the highest (26.55 °C) temperature values and the broilers fed the control diet and exposed to challenge with coccidia (PC) having the lowest values (24.89 °C). Although initial and ultimate pH values did not differ significantly (*p* > 0.05) between cinnamon-treated groups, pH decreased 24 h postmortem with increasing CBP content. Birds receiving 4 g of CBP and NC had higher initial color lightness than those receiving 66 mg of salinomycin and NC (*p* = 0.038) but were similar to those receiving 2 g of CBP, 6 g of CBP, and in the PC group. Birds that received 66 mg of salinomycin had lower final color lightness than those in the NC group and 6 g of CBP (*p* < 0.01) but were similar to those receiving 2 g of CBP, 4 g of CBP, and in the PC group.

The birds that received 66 mg of salinomycin had lower final color lightness than those in the NC group and fed 6 g of CBP (*p* < 0.01) but were similar to those receiving 2 g of CBP, 4 g of CBP, and in the PC group. Generally, the pH and color measurements of the samples were similar across all treatment groups tested, with the exception of the lightness and WI.

The higher initial lightness results in the control challenged group (PC) resulted in a lower color change and a higher WI compared with the control unchallenged group (NC), and vice versa at the ultimate lightness estimate. These findings may be attributed to the PC group having a higher pH decline than the NC group, indicating that the emeriosis challenge might alter the appearance of the lightness of breast samples. The birds that received cinnamon had lower final yellowness than those in the controls and salinomycin group (*p* < 0.021).

### 3.4. Water-Holding Capacity, Dripping Loss, Cooking Loss, Myofibril Fragmentation Index, and Shear Force

The broiler breast quality, WHC, DL, CL, MFI, and SF, of the samples at 34 d of age (14 days post-inoculation) are presented in Table 5. The CL, MFI, and SF values of the meat samples differed (*p* < 0.05) between groups, while the WHC and DL values did not differ (*p* > 0.05). The most favorable numerical values (*p* > 0.05) for WHC were obtained for the broilers fed 2 g of CBP/kg. Broilers fed 4 g of CBP/kg had the highest and worst CL value (36.64%; *p* < 0.05). In contrast, those fed 2 g of CBP/kg, the NC group, and the PC group had the lowest and the most convenient CL value (26.64%, 23.36%, and 22.46%, respectively) but were similar to those receiving 66 mg of salinomycin/kg and 6 g of CBP/kg (35.75% and 32.97%, respectively). The CL results demonstrate that breasts from chickens that received the anticoccidial (salinomycin) and 4 and 6 g of CBP shrank more when cooked than those given 2 g of CBP/kg and the control treatments (*p* < 0.001).

The broilers fed CBP had the lowest MFI value (*p* < 0.05), with 2 g of CBP/kg being the most convenient compared with the PC, NC, and salinomycin groups. The CBP treatments had higher SF values (*p* < 0.05) than the salinomycin and control groups, indicating tough meat. However, the 2 g of CBP/kg group had the lowest SF (1.69 kgf) among the cinnamon groups, indicating that it was the CBP group with the most tenderness.

### 3.5. Texture Analysis

TPA quality of breast samples from broilers fed with CBP inoculation and exposed to *E. tenella* challenge is revealed in Table 6. There were significant differences between the experimental groups in the values for hardness, springiness, and chewiness (*p* < 0.05). However, there was a tendency for the cohesiveness value to differ (*p* = 0.04) among treatments. TPA values were found to be higher (*p* < 0.05) in the breast meat of the broiler chickens receiving 66 mg salinomycin/kg. Higher hardness values (*p* < 0.05) of 2 g CBP/kg compared with 4 g and 6 g of CBP/kg groups indicated that hardness improved in broilers fed diets containing 2 g CBP/kg. Springiness and chewiness did not differ significantly among groups with different levels of CBP (*p* > 0.05). Springiness values were lower in CBP groups compared with controls and salinomycin groups (*p* < 0.0001). 

In this trial, texture profiles differed significantly between treatments, with the CBP groups having lower texture scores compared with the salinomycin treatment, but not from the control groups. However, the CBP groups had lower springiness than the control groups. The NC and 6 g CBP/kg treatments exhibited the lowest values for hardness (0.51) and for chewiness (2.06), while the salinomycin treatment demonstrated the highest values for hardness (0.94) and for chewiness (3.07).

## 4. Discussion

This study assessed the success of different levels of cinnamon as a natural herb compared with the standard synthetic anticoccidial product (salinomycin) in broilers exposed to experimentally induced challenge of coccidiosis. Few or no studies have reported the effect on the performance, carcass traits, and breast quality of broilers supplemented with CBP under coccidial challenge. The null hypothesis states that the effects of CBP on growth performance, carcass traits, and meat quality of broiler breasts are the same as the effects of the control groups (non-CBP), which is based on the *p*-value is calculated using a probability level of α = 0.05. The alternative hypothesis is that CBP inhibits the weight loss in the breast and carcass caused by *E. tenella*, which is consistent with the results of [23] who found that CBP extract inhibits the weight loss caused by *Eimeria* infection.

Several studies have evaluated cinnamon powder supplementation at different levels (g/kg) in feed: 2 [37], 10 and 20 [38], 10, 30, and 50 [39], and 30, 50, and 70 [40]. The above studies have had significant effects of the levels of cinnamon powder evaluated in broiler chickens, such as BW, FI, and FCR. The authors in [41] reported that cinnamon at different levels (2.5, 5.0, or 7.5 g) did not show any positive impact on the performance or performance index, as opposed to the results of the current study.

As expected, the results of this study showed that BW, weight gain, and feed efficiency were most adversely affected by the positive control (coccidiosis-exposed group, not treated with any natural or synthetic drug). The effect of salinomycin sodium was similar to that of the cinnamon groups as a natural herb in the 1st week after the challenge. Luckily, cinnamon outcomes, particularly cinnamon at level 2 g per kilogram of diet, have been the best during the 2nd week and entire period after challenge since BWG, FCR, and PEF were improved compared with infected groups. The exploration of alternatives to synthetic antibacterial and anti-coccidial drugs is an interesting field of research for poultry scientists [42]. The performance of birds was recovered closer to the negative control and drug-treated groups at the 2nd week following a coccidiosis challenge, which is encouraging; fortunately, the effect was exceeded in the 2g CBP group and was equal in other CBP groups to salinomycin as the synthetic drug. The positive effects of the anticoccidial cinnamon herb have been linked to the presence of active compounds present in plants that reduce the parasitic oocyst, modulate intestinal microflora, improve immunity and antioxidant status, and reduce intestinal inflammation [13,43,44,45]. Herbal anticoccidial agents maintain the growth of the broilers by reducing the destructive effects of coccidiosis [46]. The harmful effect of coccidiosis is clear in the infected group (PC), which has been improved in cinnamon powder-treated birds. Confirming our document, prior literature have also reported improved performance in response to anticoccidials and the ability of natural products to decrease the induced weight loss due to infection of birds [24,42,47]. Cinnamon bark oil up to 0.05% has a better preservative impact on the quality of lamb meat during storage it is supplemented at levels 0.01, 0.25, 0.5, and 5 g in the diet [48]. Cinnamon and/or citral supplementation in feed improved growth performance of chicken vaccinated or not vaccinated against coccidiosis to the level comparable with bacitracin and alter cecal microbiota composition [13]. The addition of cinnamon powder up to 5% appears to have no noticeable impact on growth and carcass characteristics except for the spleen and heart weight percentage [39]. However, they have discovered that it can be used to enhance cellular immune responses in broiler chicks. 

Conventional coccidiostats have been associated with undesirable residues in meat in some areas since the late 1990s [49]. However, there is no credible scientific proof to boost the allegation that residues exist or have caused issues for consumers. There is no proof of residual effects from salinomycin or other coccidiostats, with the exception of diclazuril [50]. According to [50], any coccidiostat residue in poultry meat poses a low direct hazard to human health (<1%). Moreover, none of the natural ingredients have already been studied to determine whether they also produce residues. Furthermore, the impacts of CBP on meat quality, carcass characteristics, and marketing growth of broilers infected with *E. tenella* oocysts were studied at different doses and with different results. As a follow-up to a previously published study on the effects of CBP as a coccidiosis prevention product, the effects of CBP on meat quality, growth performance, and carcass characteristics of broiler chickens infected with *E. tenella* were investigated here. Thus, this study examined the efficacy of different quantities of CBP as a natural herb compared with the current synthetic anticoccidial product (salinomycin) in birds exposed to an experimentally induced coccidial challenge. Previously, there was limited research on the effects of CBP on the carcass traits and breast quality of birds facing a coccidial challenge. 

The bird became infected with *E. tenella* after ingesting sporulated oocysts which penetrated and damaged the intestinal epithelium of the caecum due to endogenous and exogenous multiplication of the *E. tenella* stage, whereupon the developed oocysts were excreted in the feces [51]. In order to reduce the cost of eliminating the damage caused by *E. tenella* in poultry farms, the authors are actively developing various prevention methods for monitoring *E. tenella* [24,52]. The effects of *E. tenella* infection range from localized intestinal tissue degeneration to death in the most severe instances [53]. To minimize detrimental impacts on high-quality broiler meat production, new agents with cheap costs and minimum adverse effects against *E. tenella* are needed. 

Here, the positive impact of cinnamon additives, particularly at level 4 g, was obvious on most carcass traits such as the CW, carcass yield, as well as breast, heart, proventriculus, gizzard, and pancreas percentage relative to CW, increased as compared with the PC. This result may be attributed to that cinnamon can improve the digestive system due to the presence of cinnamon’s active components, such as cinnamaldehyde, which can stimulate appetite and digestion [54]. Therefore, birds who received a cinnamon diet, particularly at level 6 g, had a higher breast weight than those in the PC and similar to those received a diet supplemented with salinomycin. Our data showed that experimental treatments did not influence some of the carcass characteristics of meat, such as the relative weights of liver, leg, and fat. These findings are in agreement with [55], who found that the carcass characteristics did not change with the addition of various cinnamon oil levels and sodium butyrate except that the cholesterol level of the muscles was lowered in broilers. In part, this agrees with [56], who found that including CBP in broiler meal had no substantial effect (*p* > 0.05) on heart, breast, gizzard, cholesterol, abdominal fat, or triglycerides. However, [57] found that broilers fed CBP had a higher dressing percentage (*p* < 0.05). The findings of [58] corroborated our findings that dietary treatment with CBP had no significant effect on lymphoid organ relative weights (*p* > 0.05). Immune organ weight was significantly higher in the 5.0% CBP group, according to [40].

As expected, the results of this study showed that slaughter weight, carcass weight, and then dressing or carcass yield were adversely affected by the positive control compared with medicated or NC groups. Our findings support those of [59,60], who were orally challenged with Eimeria-populated oocysts harmed performance indices and carcass yield. Where [59] observed *Rumex nervosus* leaves can mitigate *E. tenella* suffering and improve dressing percentage. In addition, [60] observed that the dressing percentage was significantly higher in the group receiving a mixture of all 3 herbs (*Aloe barbadensis*, *Ferulafoetida regal*, and *Tamarindus indica*) at 2 mL/L mixed with citric acid and lowest in the control group (without medicinal herbs supplementation and challenged with Eimeria-populated oocysts).

Instead of glucose under aerobic conditions, muscle glycogen is the primary metabolic fuel for anaerobic glycolysis after slaughter. When anaerobic glycolysis occurs, pyruvate is reduced to lactate. The accumulation of lactate causes a decrease in pH whenever the muscle is converted to meat [61]. When the pH reaches acidic conditions, glycolytic enzymes are probably inactivated. Thus, [62] mention that pH is one of the most important alterations that occur during rigor mortis and that it has a direct influence on the quality characteristics of the meat, such as juiciness, texture (tenderness), WHC, color, and shelf life. The meat of birds with a high pH has a higher WHC than meat with a lower pH. The pH of meat is easily determined by its color. The pH of meat is high when it is very dark and low when it is very light. The lower pH in bird meat groups with herbs may be responsible for inhibiting the integration of the deterioration of the growth of microorganisms [63]. The meat quality (PSE, DFD) of birds can be measured quickly and precisely to determine the pHi value of meat samples. The threshold pH value categories of the breast meat of broilers are 5.8 (reddish, soft, and exudative), 5.9–6.2 (standard meat quality), and 6.3 (pale, firm, and non-exudative or dark, firm, and dry) [64]. A duration of 15 min postmortem, the pH parameters were a good predictor of meat traits [65] and ranged from 5.78 to 6.59 [66]. The authors of [61] reviewed the literature and found that the highest quality commercial poultry meat products are more likely to fall within the pH range of 5.7–6.0. In this study, the initial pH ranged from 6.08 to 6.36, and the ultimate pH ranged from 5.71 to 5.83. There was acceptable color and increased wateriness (reddish, soft, and exudative) as well as pale color and good juice retention (pale, firm, and non-exudative). Although there were no significant differences between the experimental groups, the pH of chicken breast meat in the present study decreased mathematically with a rise in CBP doses. The variation in the pH could be attributed to the high antioxidant of cinnamon and its other hydroxyl derivatives attributed to the action of hydroxyl radicals (^•^OH) present in the phenyl ring of phenolic compounds acting as hydrogen donors [67]. Therefore, increasing CBP nutritional levels in diets increase the donation of hydroxyl groups, effectively reducing the pH value noticed in broiler breast muscle.

Meat color is influenced by many influences, such as pre-slaughter factors, stunning methods, cooling regimes, moisture content, heme pigments, protein physical status, strain, stress, and sex [4,5]. In [68], it was pointed out that raising the L* value was desirable in terms of consumer acceptance. Here, the 4 g of CBP/kg group had a higher initial L* value and then a higher WI and lower color change, while the ultimate L* value and WI were increased with increasing doses of CBP. Metmyoglobin cumulating on the exterior part of the storage meat contributes to the discoloration of the meat [69], which eliminates the a* value discrepancies. The formation of metmyoglobin and an elevation of lipid oxidation are the key components responsible for distinctions in the b* value [70]. Differences in initial and ultimate of L* values, total color change (∆E), and WI were found by [31] when testing CBP. Few researchers have described ∆E and BI parameters, which could assist in elucidating the L*, a*, and b* behavior in bird breast meat, as impacted by CBP intake in the feed. These findings revealed that partial alterations in color variables could be due to the effects of experimental groups on the ∆E and WI. However, other authors, such as those of [55], did not discover any effect of CBP supplemented diets on broiler meat color measurements. Thus, decreasing water retention tends to lead to less reflective surface light that reduces L* values [71], and is associated with a decrease in the nutritional value of the meat due to the loss of some nutrients, and as a result, the breast meat becomes less tender. The experimental treatment had no effect on both WHC and DL (*p* > 0.05). WHC is a phrase used to refer to a muscle’s ability to bind water under a particular set of circumstances. Commonly, the increase in muscle fat content results in higher WHC and a reduction in the percentage CL [72]. After death, oxygen deprivation causes lactic acid production, leading to a decline in pH, which causes protein denaturation, loss of protein solubility, and an overall reduction in the number of reactive groups available for water binding on muscle protein [62]. CL is a measurement of how much water is lost during cooking because of shrinkage. The degree of shrinkage that occurs during cooking is proportional to the loss of juiciness on the palate. CL was considerably lower in the study when birds were fed a diet containing 2 g of CBP/kg. In contrast, [73] did not observe any influence of 0.5 or 1 mL of cinnamon oil in broiler diets on CL.

The CBP treatment affected the MFI of the breast muscle. Myofibril fragmentation refers to the degree to which homogenization causes myofibrils to be destroyed. The authors of [74] have shown that the values of MFI are strongly correlated with other muscle measurements, such as tenderness and SF. Therefore, cinnamon supplementation could cause less fragmentation of myofibrils. On another hand, the SF in the breast muscle of birds ranged from 5.5 to 5.8 kgf/g [75] and between 2.71 and 3.31 kgf/g [76]. Therefore, the CBP groups in this trial had no effect on meat tenderness as the SF values were between 1.69 and 2.10 kgf/g, and they were almost 67% and 37% lower than the values reported by [75], respectively. However, the CBP treatments had higher SF values than the controls, and they were similar to the salinomycin group. These findings are in contrast with [55], who found that different levels of diets supplemented with cinnamon oil and sodium butyrate did not influence the meat SF value (kg force/cm^2^) of the broilers.

In comparison with the salinomycin group, the CBP groups had lower levels of texture profiles. Additionally, the CBP groups had lower springiness than the control groups. Meat texture was evaluated using TPA and SF as having a myofibril structure. Recent investigations have been performed on the quality of meat or carcass characters of birds fed diets containing either powder or plant extracts [27,30,77,78,79,80]. However, little or no research has been conducted on the effect of CBP on TPA. The authors in [81] found that springiness and cohesiveness decreased as more cinnamon extract was added to sourdough bread. However, hardness, chewiness, and gumminess were reversed. It was thought that adding the cinnamon extract to a bread recipe would be beneficial. In our experiment, the hardness, springiness, cohesiveness, and chewiness decreased in CBP treatments compared with the salinomycin treatment, but there was no difference between the control groups.

Despite the fact that the supplemented CBP resulted in conflicting patterns in terms of SF and TPA, the treatment without CBP (NC) produced the best SF and hardness values. This was also the case for cohesiveness and springiness. The addition of CBP to broiler diets resulted in an increase in meat toughness in general. As a result, adding natural antioxidant compounds to meat can improve its quality, and cinnamon has the highest antioxidant capacity due to its high phenolic content [20,82]. Natural antioxidants, on the other hand, have been found to have little or no effect on the sensory characteristics of meat by some researchers. Supplementation with cinnamon oil, for example, had no effect on the quality of chicken meat, according to [55].

## 5. Conclusions

In summary, *Eimeria tenella* infection has a negative impact on growth performance, slaughter weight, carcass yield, and most carcass characteristics of broiler chicken; on the other hand, the use of cinnamon as alternatives to anticoccidials and ionophore coccidiostats, can mitigate these effects. Moreover, the addition of cinnamon was able to improve some physicochemical properties without affecting the meat’s quality. However, in the breasts of birds given cinnamon, MFI decreased and toughness increased, when compared with the other experimental groups. Although this is a consistent experimental paradigm that is highly applicable to commercial conditions, more research into the use of cinnamon to improve the meat quality and productivity of broiler chickens in both healthy and sick conditions is needed.

## Figures and Tables

**Table 1 animals-12-00166-t001:** Live body weight (BW), average live body gain (BWG), average feed intake (FI), feed conversion ratio (FCR), and production efficiency factor (PEF) of broiler chickens given experimental diets (cinnamon), post coccidial challenge period (0–14 dpi).

				Cinnamon (g/kg)			
Treatment ^1^	NC	PC	Salinomycin	2	4	6	*p*-Value
1 to 7 dpi							
BW (kg)	1.273 ± 0.01 ^2a^	1.122 ± 0.01 ^c^	1.238 ± 0.03 ^ab^	1.178 ± 0.03 ^abc^	1.161 ± 0.02 ^bc^	1.228 ± 0.02 ^abc^	0.043
BWG (g)	82.91 ± 1.77 ^a^	56.45 ± 1.86 ^b^	57.75 ± 0.78 ^b^	54.28 ± 1.38 ^b^	61.03 ± 1.96 ^b^	64.45 ± 1.96 ^b^	0.001
FI (g)	117.2 ± 2.00	99.6 ± 2.06	104.4 ± 1.82	99.5 ± 1.88	111.2 ± 8.28	102.3 ± 5.52	0.131
FCR (g:g)	1.41 ± 0.02 ^b^	1.77 ± 0.06 ^a^	1.81 ± 0.04 ^a^	1.84 ± 0.05 ^a^	1.83 ± 0.04 ^a^	1.59 ± 0.05 ^ab^	0.055
PEF	336.2 ± 11.4 ^a^	230.6 ± 12.2^b^	254.2 ± 7.1 ^b^	239.3 ± 11.0 ^b^	237.2 ± 15.4 ^b^	289.5 ± 14.2 ^ab^	0.008
8 to 14 dpi							
BW (kg)	1.870 ± 0.03	1.649 ± 0.02	1.850 ± 0.03	1.858 ± 0.01	1.752 ± 0.02	1.737 ± 0.03	0.067
BWG (g)	85.23 ± 3.34^b^	75.58 ± 2.68 ^c^	87.25 ± 2.26 ^b^	97.08 ± 0.81 ^a^	84.36 ± 4.11 ^b^	72.76 ± 3.08^c^	0.007
FI (g)	134.6 ± 2.78	130.2 ± 2.27	130.7 ± 3.85	126.6 ± 1.75	133.7 ± 2.01	125.3 ± 1.91	0.095
FCR (g:g)	1.58 ± 0.05 ^ab^	1.73 ± 0.05 ^a^	1.51 ± 0.04 ^ab^	1.31 ± 0.02 ^b^	1.60 ± 0.06 ^ab^	1.73 ± 0.06 ^a^	0.043
PEF	438.8 ± 14.0 ^ab^	354.2 ± 12.6 ^b^	456.8 ± 16.5 ^ab^	537.1 ± 13.3 ^a^	411.4 ± 17.7 ^b^	375.6 ± 15.5 ^b^	0.038
1 to 14 dpi							
BWG (g)	84.07 ± 1.81 ^a^	66.02 ± 1.89 ^d^	72.50 ± 0.86 ^bc^	75.68 ± 0.86 ^b^	72.70 ± 1.62 ^bc^	68.60 ± 1.50 ^cd^	<0.001
FI (g)	125.88 ± 2.16	114.9 ± 1.88	117.52 ± 2.20	113.07 ± 1.13	122.43 ± 0.70	113.78 ± 1.46	0.148
FCR (g:g)	1.50 ± 0.02 ^c^	1.74 ± 0.05 ^a^	1.62 ± 0.03 ^b^	1.50 ± 0.02 ^c^	1.69 ± 0.03 ^ab^	1.66 ± 0.04 ^ab^	0.001
PEF	387.5 ± 9.1 ^a^	292.4 ± 12.1 ^c^	355.5 ± 8.0 ^ab^	388.2 ± 8.5 ^a^	324.3 ± 16.0 ^bc^	332.6 ± 12.1 ^b^	<0.001

^1^ Treatments: NC—negative control, unsupplemented, unchallenged; PC—positive control, unsupplemented, challenged; Salinomycin—basal diet supplemented with coccidiostat salinomycin, challenged; Cinnamon—groups whose basal diet was supplemented with 2, 4, and 6 g cinnamon powder/kg diet, respectively, challenged. ^a–d^ Different letters indicate statistically significant differences (*p* < 0.05). ^2^ Values are presented in means ± standard error (SE) (*n* = 5).

**Table 2 animals-12-00166-t002:** Carcass variables of broiler chickens supplemented with cinnamon powder, 14 days post-infection.

				Cinnamon (g/kg)			Probability
Treatment ^1^	NC	PC	Salinomycin	2	4	6	
Live wt (kg)	1.779 ± 32.3 ^2a^	1.505 ± 37.4 ^c^	1.736 ± 24.2 ^a^	1.872 ± 41.7 ^a^	1.810 ± 29.5 ^a^	1.791 ± 32.0 ^a^	<0.0001
Carcass wt (kg)	1.194 ± 19.6 ^a^	0.974 ± 24.9 ^b^	1.179 ± 18.9 ^a^	1.245 ± 28.0 ^a^	1.222 ± 24.7 ^a^	1.204 ± 27.6 ^a^	<0.0001
CY% ^3^	67.14 ± 0.33 ^a^	64.72 ± 0.15 ^b^	67.88 ± 0.39 ^a^	66.53 ± 0.73 ^a^	67.78 ± 0.63 ^a^	67.19 ± 0.42 ^a^	0.003
Heart	0.44 ± 0.01 ^b^	0.45 ± 0.01 ^b^	0.46 ± 0.01 ^b^	0.54 ± 0.01 ^a^	0.54 ± 0.02 ^a^	0.46 ± 0.002 ^b^	<0.0001
Liver	1.82 ± 0.08	1.95 ± 0.06	2.00 ± 0.08	1.84 ± 0.02	1.75 ± 0.12	1.96 ± 0.01	0.133
Proventriculus	0.37 ± 0.02 ^bc^	0.39 ± 0.02 ^bc^	0.34 ± 0.04^c^	0.41 ± 0.03 ^bc^	0.53 ± 0.03 ^a^	0.44 ± 0.03 ^b^	0.003
Gizzard	1.96 ± 0.08 ^c^	2.35 ± 0.09 ^b^	2.00 ± 0.15^c^	2.55 ± 0.11 ^ab^	2.69 ± 0.04 ^a^	2.63 ± 0.08 ^ab^	<0.0001
Bursa	0.19 ± 0.03	0.22 ± 0.03	0.15 ± 0.03	0.21 ± 0.01	0.25 ± 0.01	0.21 ± 0.01	0.115
Spleen	0.09 ± 0.002	0.10 ± 0.001	0.11 ± 0.01	0.07 ± 0.002	0.10 ± 0.01	0.09 ± 0.01	0.267
Thymus	0.27 ± 0.02	0.32 ± 0.02	0.33 ± 0.02	0.35 ± 0.05	0.40 ± 0.06	0.43 ± 0.03	0.094
Breast	26.87 ± 0.53 ^ab^	25.64 ± 0.14 ^b^	27.71 ± 0.60 ^ab^	26.14 ± 0.19 ^b^	26.77 ± 0.44 ^ab^	28.69 ± 0.74 ^a^	0.051
Leg	19.14 ± 0.48	20.17 ± 0.35	19.40 ± 0. 0.36	20.21 ± 0.53	19.92 ± 0.22	19.53 ± 0.21	0.275
Fat	0.80 ± 0.14	0.60 ± 0.08	0.85 ± 0.17	0.78 ± 0.03	0.57 ± 0.16	0.55 ± 0.08	0.274
Pancreas	0.307 ± 0.03 ^c^	0.389 ± 0.02 ^b^	0.418 ± 0.03 ^ab^	0.417 ± 0.02 ^ab^	0.474 ± 0.03 ^a^	0.403 ± 0.02 ^ab^	0.006

^1^ Treatments: NC—negative control, unsupplemented, unchallenged; PC—positive control, unsupplemented, challenged; Salinomycin—basal diet supplemented with coccidiostat salinomycin, challenged; Cinnamon—groups whose basal diet supplemented with 2, 4, and 6 g of cinnamon powder/kg of diet, respectively, challenged. ^2^ The data is presented as means with standard errors (*n* = 5). ^a–c^ Means in the rows with different superscripts differ significantly (*p* < 0.05), ^3^ Dressing percentage or carcass yield (CY%) = (carcass weight/Live weight) * 100.

**Table 3 animals-12-00166-t003:** Core temperature, pH, and color of the pectoralis major at 34 d of age in broilers fed diets containing varying amounts of cinnamon bark powder (CBP) were measured 1 h postmortem.

				Cinnamon (g/kg)			Probability
Treatment ^1^	NC	PC	Salinomycin	2	4	6	
Core Temperature	25.5 ± 0.15 ^2cd^	24.89 ± 0.23 ^d^	25.78 ± 0.23 ^bc^	26.55 ± 0.10 ^a^	25.6 ± 0.19 ^c^	26.41 ± 0.05 ^ab^	<0.0001
Initial pH	6.08 ± 0.07	6.16 ± 0.08	6.23 ± 0.02	6.25 ± 0.10	6.16 ± 0.09	6.36 ± 0.04	0.104
L1	42.12 ± 0.94 ^b^	45.29 ± 0.74 ^a^	42.12 ± 0.29 ^b^	44.10 ± 0.59 ^ab^	45.35 ± 0.60 ^a^	44.48 ± 0.47 ^ab^	0.002
a1	6.59 ± 1.04	5.43 ± 0.81	5.38 ± 0.52	5.88 ± 0.57	6.48 ± 0.38	6.25 ± 0.45	0.695
b1	7.7 ± 0.34	7.36 ± 0.27	5.96 ± 0.39	6.32 ± 0.25	7.5 ± 0.87	6.91 ± 0.48	0.104
∆E	52.71 ± 1.05 ^a^	49.4 ± 0.73 ^b^	52.44 ± 0.31 ^a^	50.55 ± 0.61 ^ab^	49.49 ± 0.54 ^b^	50.25 ± 0.50 ^ab^	0.004
Hue angle (◦)	50.86 ± 4.54	54.39 ± 4.25	48.31 ± 1.31	47.62 ± 2.00	48.28 ± 2.60	47.86 ± 2.04	0.589
Saturation index	10.28 ± 0.75	9.27 ± 0.52	8.03 ± 0.62	8.65 ± 0.55	9.95 ± 0.86	9.34 ± 0.58	0.205
Browning index	31.5 ± 2.91	26.11 ± 1.29	24.26 ± 1.96	24.88 ± 1.65	28.11 ± 2.56	26.88 ± 1.98	0.223
Whiteness index	41.20 ± 1.04 ^b^	44.49 ± 0.71 ^a^	41.55 ± 0.32 ^b^	43.43 ± 0.61 ^ab^	44.41 ± 0.50 ^a^	43.69 ± 0.52 ^ab^	0.004

^1^ Treatments: NC—negative control, unsupplemented, unchallenged; PC—positive control, unsupplemented, challenged; Salinomycin—basal diet supplemented with coccidiostat salinomycin, challenged; Cinnamon—groups whose basal diet supplemented with 2, 4, and 6 g cinnamon powder/kg diet, respectively, challenged. ^2^ Each mean is based on measurements from 5 birds per treatment. ^a–d^ Means in the same rows with different superscripts differ significantly at *p* < 0.05; L1, a1, and b1—initial lightness, redness, and yellowness, respectively; ∆E—total color change.

**Table 4 animals-12-00166-t004:** pH, and color of the pectoralis major at 34 d of age in broilers fed diets containing varying amounts of cinnamon bark powder (CBP) were measured 24 h postmortem.

				Cinnamon (g/kg)			Probability
Treatment ^1^	NC	PC	Salinomycin	2	4	6	
Ultimate pH	5.78 ± 0.02 ^2^	5.79 ± 0.02	5.76 ± 0.02	5.83 ± 0.04	5.77 ± 0.02	5.71 ± 0.03	0.090
pH decline	0.31 ± 0.08	0.37 ± 0.08	0.46 ± 0.03	0.42 ± 0.07	0.39 ± 0.05	0.65 ± 0.06	0.067
L2	49.91 ± 0.26 ^a^	44.29 ± 1.01 ^bc^	43.18 ± 0.77 ^c^	45.46 ± 0.32 ^bc^	46.26 ± 0.88 ^bc^	47.47 ± 1.31 ^ab^	0.0001
a2	7.25 ± 0.89	7.94 ± 0.53	7.55 ± 0.87	7.61 ± 0.53	8.03 ± 0.16	8.54 ± 1.14	0.873
b2	10.48 ± 0.70 ^bc^	10.79 ± 0.85 ^bc^	9.95 ± 0.51 ^c^	13.22 ± 0.80 ^a^	11.97 ± 0.63 ^abc^	12.17 ± 0.48 ^ab^	0.021
∆E	45.48 ± 0.30 ^c^	51.09 ± 1.10 ^ab^	52.01 ± 0.85 ^a^	50.28 ± 0.36 ^ab^	49.33 ± 0.90 ^ab^	48.38 ± 1.11 ^bc^	0.004
Hue angle (◦)	55.68 ± 2.73	53.49 ± 0.73	53.44 ± 2.36	60.12 ± 0.92	55.94 ± 1.00	55.4 ± 4.34	0.589
Saturation index	12.81 ± 0.95	13.4 ± 0.99	12.54 ± 0.87	15.26 ± 0.93	14.42 ± 0.61	15.09 ± 0.41	0.205
Browning index	33.82 ± 2.60	41.13 ± 3.96	38.71 ± 3.16	46.25 ± 3.13	42.44 ± 2.52	42.52 ± 1.10	0.223
Whiteness index	48.25 ± 0.29 ^a^	42.66 ± 1.12 ^b^	41.79 ± 0.84 ^b^	43.33 ± 0.38 ^b^	44.34 ± 0.92 ^b^	45.31 ± 1.17 ^ab^	0.004

Ultimate pH and color values of the breast muscle were measured. ^1^ Treatments: NC—negative control, un challenged, not supplemented; PC—positive control, challenged, not supplemented; Salinomycin—challenged, basic diet supplemented with salinomycin; Cinnamon—groups whose basal diet supplemented with 2, 4, and 6 g CBP/kg diet, respectively, challenged. ^2^ Each mean is calculated using data from 5 birds per group with standard errors. ^abc^ Means differ significantly in the same rows with different superscripts at *p* < 0.05; abbreviations: L2, a2, and b2, final lightness; redness; and yellowness, respectively; ∆E—total color change.

**Table 5 animals-12-00166-t005:** Water-holding capacity (WHC), dripping loss (DL), cooking loss (CL), myofibril fragmentation index (MFI), and shear force (SF) in broilers fed diets containing varying amounts of cinnamon bark powder (CBP).

				Cinnamon (g/kg)			*p*-Value
Parameter Treatment ^1^	NC	PC	Salinomycin	2	4	6	
WHC	31.85 ± 1.23 ^2^	32.33 ± 0.70	30.38 ± 0.48	28.8 ± 1.10	31.92 ± 0.65	27.90 ± 0.82	0.0715
DL	1.55 ± 0.06	0.99 ± 0.07	1.01 ± 0.05	1.31 ± 0.10	1.45 ± 0.07	1.25 ± 0.09	0.1026
CL	23.35 ± 0.86 ^b^	22.45 ± 0.95 ^b^	35.75 ± 1.18 ^a^	26.65 ± 1.19 ^b^	36.63 ± 0.85 ^a^	32.98 ± 1.06 ^a^	<0.0001
MFI	112.85 ± 5.51 ^a^	104.80 ± 6.30 ^a^	123.33 ± 6.12 ^a^	64.88 ± 3.28 ^b^	73.38 ± 5.59 ^b^	67.35 ± 4.78 ^b^	<0.0001
SF (kgf)	1.33 ± 0.05 ^b^	1.40 ± 0.05 ^b^	1.47 ± 0.06 ^b^	1.69 ± 0.04 ^ab^	2.10 ± 0.09 ^a^	1.96 ± 0.08 ^a^	0.0002

^1^ Treatments: NC—negative control, unsupplemented, unchallenged; PC—positive control, unsupplemented, challenged; Salinomycin—basal diet supplemented with coccidiostat salinomycin, challenged); Cinnamon—groups whose basal diet supplemented with 2, 4, and 6 g of cinnamon powder/kg of diet, respectively, challenged. ^2^ The data is presented as means of 5 birds per treatment with standard errors of each mean (*n* = 5). ^abc^ Means in the same rows with different superscripts differ significantly at *p* < 0.05.

**Table 6 animals-12-00166-t006:** Texture profile in broilers fed diets containing varying amounts of cinnamon bark powder (CBP).

				Cinnamon (g/kg)			*p*-Value
Treatments ^1^	NC	PC	Salinomycin	2	4	6	
Hardness (kg)	0.51 ± 0.02 ^2b^	0.62 ± 0.03 ^b^	0.94 ± 0.02 ^a^	0.73 ± 0.02 ^ab^	0.67 ± 0.02 ^b^	0.67 ± 0.02 ^b^	<0.0001
Springiness (mm)	0.91 ± 0.01 ^a^	0.88 ± 0.02 ^a^	0.90 ± 0.01 ^a^	0.74 ± 0.01 ^b^	0.75 ± 0.01 ^b^	0.79 ± 0.02 ^b^	<0.0001
Cohesiveness (dimensionless)	0.43 ± 0.01 ^ab^	0.44 ± 0.01 ^ab^	0.45 ± 0.01 ^a^	0.40 ± 0.01 ^b^	0.41 ± 0.01 ^ab^	0.41 ± 0.01 ^b^	<0.0001
Chewiness (g mm)	2.07 ± 0.05 ^b^	2.46 ± 0.05 ^b^	3.67 ± 0.03 ^a^	2.10 ± 0.07 ^b^	2.11 ± 0.07 ^b^	2.06 ± 0.06 ^b^	<0.0001

^1^ Treatments: NC—negative control, unsupplemented, unchallenged; PC—positive control, unsupplemented, challenged; Salinomycin—basal diet supplemented with *coccidiostat* salinomycin, challenged); Cinnamon—groups whose basal diet supplemented with 2, 4, and 6 g of cinnamon powder/kg of diet, respectively, challenged). ^2^ Each mean is based on measurements of 5 birds per treatment with standard errors of each mean (mean ± SE) (*n* = 5) at the end of the trial. ^a^ and ^b^ Means in the same row with different superscripts differ significantly at *p* < 0.05.

## Data Availability

All data sets collected and analyzed during the current study are available from the corresponding author on fair request.

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
