# Peer review of "Dietary Cinnamon Bark Affects Growth Performance, Carcass Characteristics, and Breast Meat Quality in Broiler Infected with Eimeria tenella Oocysts"

_animals, 2022, doi:10.3390/ani12020166_

Round 1
Reviewer 1 Report
The manuscript described on effects of Cinnamon Bark on meat quality is interesting, but there are some major concerns as bellows.
Major points
- Each group has five chicken samples (n=5). Without repeat works, it is difficult to provide high fidelity to the data.
- All tables included only one SEM. However, each value requires each SEM for statistical analysis. Please add SEM to each value.
Minor points
-In line 191, CBP and Sacox may be CBP or Sacox.
-In line 308, the sentence “There were significant variation in slaughter variables was found between experimental groups” looks incomplete.
-In line 411, It is difficult to understand the sentences “The effects of cinnamon bark on meat quality and carcass characteristics were not investigated directly in this experiment, but rather the effects of the herbs as a remedy on meat quality and carcass characteristics during infection with E. tenella.” Please check them.
-In line 441, It is difficult to understand the sentences “This may be attributed to Cinnamon can be able to make changes in the digestive system”
- In line 453-461, the complicated sentences needed to be rearranged.
Author Response
Comments and Suggestions for Authors ( Reviewer 1):
Thanks for your effort in reviewing our manuscript, the authors have been provided responses requested based on your reviewing comments as following:
The manuscript described on effects of Cinnamon Bark on meat quality is interesting, but there are some major concerns as bellows.
Major points
Comment
- Each group has five chicken samples (n=5). Without repeat works, it is difficult to provide high fidelity to the data.
Response
Thank you so much for your feedback. The authors conducted these trial and predicted that five birds per treatment would be sufficient for carcass traits and meat quality (one male bird per replicate was randomly selected for slaughtered). Six birds per treatment were used by Liu et al. (2019) and A AL-Sagan et al. (2020) (one bird per replicate was randomly selected for slaughter). The biological samples for SF and TPA tests were five, but the physical replicate was 5 for each biological sample, so the total samples will be 25. The biological samples for color and pH measurements were five, but the physical replicate was 3 for each biological sample, so the total samples will be 15. The biological samples for water holding capacity and MFI measurements were five, but the physical replicate was 2 for each biological sample, making the total samples 10. We've now included performance data from 30 replicates (5 birds per replicate), with 25 birds in each treatment.
- All tables included only one SEM. However, each value requires each SEM for statistical analysis. Please add SEM to each value.
Response
Done as requested. Thank you.
Minor points
Comment
-In line 191, CBP and Sacox may be CBP or Sacox.
Response
CBP or Sacox, corrected by replaced “and” by “or”
Comment
-In line 308, the sentence “There were significant variation in slaughter variables was found between experimental groups” looks incomplete.
Response
The incomplete sentence has now been completed and highlighted in yellow:
“Except for liver, leg, fat and lymphoid organs (bursa, thymus, and spleen) values, there was significant variation in slaughter variables between treatments.”
Comment
-In line 411, It is difficult to understand the sentence “The effects of cinnamon bark on meat quality and carcass characteristics were not investigated directly in this experiment, but rather the effects of the herbs as a remedy on meat quality and carcass characteristics during infection with E. tenella.” Please check them.
Response
Replaced by “As a follow-up to a previously published study on the effects of CBP as a coccidiosis prevention product, the effects of CBP on meat quality, growth performance, and carcass characteristics of broiler chickens infected with E. tenella were investigated here”.
Comment
-In line 441, It is difficult to understand the sentences “This may be attributed to Cinnamon can be able to make changes in the digestive system”
Response
The sentence that follows has been rewritten to make it more understandable: “This could be attributed to Cinnamon's active components, such as cinnamaldehyde, which can stimulate appetite and digestion (Kumar et al., 2014)”.
Comment
- In lines 453-461, the complicated sentences needed to be rearranged.
Response
The complex sentences have been rearranged as follows: “In part, this agrees with Shirzadegan (2014), who found that including CBP in broiler meal had no substantial effect (P > 0.05) on heart, breast, gizzard, , cholesterol, abdominal fat, or triglycerides. However, Eltazi (2014) found that broilers fed CBP had a higher dressing percentage (P < 0.05). The findings of Najafi and Taherpour (2014), corroborated our findings that dietary treatment with CBP had no significant effect on lymphoid organ relative weights (P > 0.05). Immune organ weight was significantly higher in the 5.0% CBP group, according to Sang-Oh et al. (2013)”.
Thanks for your effort in reviewing our manuscript.
01 Dec 2021
Reviewer 2 Report
This study investigates the effect of dietary cinnamon bark affects carcass characteristics and breast meat quality in broiler infected with Eimeria tenella oocyts. The results showed that cinnamon addition decreased MFI and increased toughness of breast meat of Eimeria-infected broilers. However, there are several crucial error the authors must revise in future.
- The authors should provide the growth performance data, especially the feed intake.
- The authors should provide the active composition of cinnamon in the Material and Methods
- The authors used the gradual increase of cinnamon dosage in the diets. Thus, the authors should analyze the trend of responsive indices with dosage.
Others:
The authors should improve English expression in whole manuscript, especially some grammar errors, which made the readers confused for your results.
Title: should be changed into Dietary cinnamon bark affects carcass characteristics and breast meat quality in broiler infected with Eimeria tenella oocyts.
ABSTRACT
Meat tenderness is one parameter of meat quality
What is coccidiosis? Do you identify in the research?
Line 43, have been should be changed into were
Line 56-57, without affecting meat quality, which are not consistent with the following description “ decreasing MFI and increasing toughness
Material and methods
Line 150, this sentence made the readers confused. Please redescribe it or delete it.
Line 154, what type of Eimeria tenella? Provide the detailed information. And please provide the simple procedure of inoculation.
Line 154, please specify the age of birds for inoculation. Why do the authors choose 21 days of age, not the 14 days of age? In addition, why do you choose the 14 dpi for sampling, not the 7 dpi?
Line 155-158. What do the two sentences mean? The same batch of broilers ?? if not, delete these sentence.
Line 174, the temperature was set at 35 ℃ for the first week, which was higher for the starter broilers in the first week. Please specify the decrease program for the temperature control in line 175.
Line 189, please provide the information of Sacox product, not showing this product name in other places.
Results
Line 298, please provide enough information including the Table to show the anti-coccidial activity of CBP.
Line 303, please provide enough information of composition of cinnamon bark extract in the Material and Methods part.
Line 308, are provided should be changed into are shown
Line 308, the sentence is not correct. There is grammar error, two verbs.
Line 310, there is a grammar error in this sentence, did not different should be changed into did not differ or there were not significant differences in
Line 437, there is error in this sentence. On the gastrointestinal tracts and nutrient absorption, which parameters represent these functions?? On the gastrointestinal tracts and nutrient absorption or on the most carcass traits ?? confused to the readers.
Line 440 delete were, changed into increased as compared to the PC
Line 441, this should be changed into this result may be attributed to that cinnamon can improve the digestive system. please provide the reference.
Tables,
I suggest the authors analyzed the trend of responsive indices to the dosage of cinnamon.
Author Response
Comments and Suggestions for Authors (Reviewer 2)
Thank you for your time and effort in reviewing our manuscript and providing us with all of your comments and suggestions.
Comment
This study investigates the effect of dietary cinnamon bark affects carcass characteristics and breast meat quality in broiler infected with Eimeria tenella oocyts. The results showed that cinnamon addition decreased MFI and increased toughness of breast meat of Eimeria-infected broilers. However, there are several crucial error the authors must revise in future.
- The authors should provide the growth performance data, especially the feed intake.
Response
The requested “growth performance data” was provided and highlighted in yellow. Thank you.
Comment
- The authors should provide the active composition of cinnamon in the Material and Methods
Response
The active composition of cinnamon were provided in “2.5. Preparation and Compositions Cinnamon Bark Powder” in the Material and Methods
“A total of 26 different active compounds with the highest quality were detected by GC-MS in the CNB extract, particularly Cinnamaldehyde, 3-phenyl-, hexadecanoic acid, (E)-2-propenal, methyl ester, 14-methyl-,methyl ester, pentadecanoic acid, oxime-, methoxy-phenyl-, and 2-methyl-benzofuran as referred previously (Qaid et al., 2021b).
Comment
- The authors used the gradual increase of cinnamon dosage in the diets. Thus, the authors should analyze the trend of responsive indices with dosage.
Response
Thank you so much for your advice. Of course, trend analysis using orthogonal polynomials is used when researchers want to study a trait's response to a change at different levels to a quantitative factor in order to determine the type of relationship (linear, quadratic, or cubic relationship) between the studied trait and the levels of treatment within multiple periods. However, in this study, authors not only used the gradual increase of cinnamon dosage in the diets but also group of infected broiler treated with Sacox, uninfected broiler group that received a commercial diet alone (NC), and infected broiler group that received a commercial diet alone (PC). Thus, Duncan's multiple range test was used to separate multiple comparisons and detect the significant differences between measurement means.
Others:
Comment
The authors should improve English expression in whole manuscript, especially some grammar errors, which made the readers confused for your results.
Response
Thank you so much for your insightful comment. To improve English expression, the authors rearranged sentences that confused readers about our findings and checked for grammar errors throughout the manuscript. The authors highlighted corrected words by yellow.
Comment
Title: should be changed into Dietary cinnamon bark affects carcass characteristics and breast meat quality in broiler infected with Eimeria tenella oocyts.
Response
Title: changed to “Dietary cinnamon bark affects growth performance, carcass characteristics, and breast meat quality in broiler infected with Eimeria tenella oocysts” as requested.
ABSTRACT
Comment
Meat tenderness is one parameter of meat quality
Response
Thank you incredibly much. Of course, meat tenderness is one criterion for meat quality. As a result, the authors changed "breast meat tenderness and quality" in the abstract to "breast meat quality".
Comment
What is coccidiosis? Do you identify in the research?
Response
To identify coccidiosis, the authors added “coccidiosis as one of a protozoan parasitic diseases” in the abstract section and for greater clarity in the introduction section of the research: “Of parasitic diseases, Coccidiosis is a protozoan disease that causes enteritis, hemorrhagic cecal lesions, and bloody diarrhoea with a significant economic losses worldwide to the poultry industry (Abouelenien et al., 2021).
Comment
Line 43, have been should be changed into were
Response
The words " have been" in Line 43 were changed into "were" as requested.
Comment
Line 56-57, without affecting meat quality, which are not consistent with the following description “ decreasing MFI and increasing toughness
Response
Thank you for bringing this to our attention. The author changed this sentence “Moreover, the addition of cinnamon was able to improve some physicochemical properties with affecting meat quality such as decreasing MFI and increasing toughness in the breasts of birds receiving cinnamon compared to the other experimental groups”. As demonstrated by the following sentence “Furthermore, when compared to the other experimental groups, the addition of cinnamon improved some physicochemical properties with some affecting meat quality, such as decreasing MFI and increasing toughness in cinnamon-treated groups”.
Material and methods
Comment
Line 150, this sentence made the readers confused. Please redescribe it or delete it.
Response
This sentence was removed to avoid confusion among readers.
Comment
Line 154, what type of Eimeria tenella? Provide the detailed information. And please provide the simple procedure of inoculation.
Response
The type of Eimeria is Eimeria tenella, However we repeated (E. tenella) as abbreviation for Eimeria tenella.
Our previous study (Qaid et al., 2021a) described the source of Eimeria tenella oocysts, the sporulation of unsporulated oocysts, the identification, passage, and propagation of sporulated oocysts, and the inoculation procedure.
Comment
Line 154, please specify the age of birds for inoculation. Why do the authors choose 21 days of age, not the 14 days of age? In addition, why do you choose the 14 dpi for sampling, not the 7 dpi?
Response
According to previous research, birds were infected with Eimeria tenella at 21 days of age (Al-Quraishy et al., 2020; Mengistu et al., 2021; Soutter et al., 2021). The authors choose the 7 dpi for anticoccidial indices (data published (Qaid et al., 2021a). For growth performance sampling, we chose 7 and 14 dpi. Following the study (Suliman et al., 2020), we chose the last day of the experiment (34 day of age “14 dpi here”) for meat quality and carcass characteristics sampling in order to mimic carcass traits of commercial poultry at marketing weights and identify meat quality.
Comment
Line 155-158. What do the two sentences mean? The same batch of broilers ?? if not, delete these sentence.
Response
Yes of course, this study is an extension of the same broiler batch experiment in which the anticoccidial indicators of CBP evaluation, namely the number of fecal oocysts, survival rate, bloody diarrhea, and lesion scores, were included. As a result, this investigation does not address the assessment of anticoccidial indicators.
Comment
Line 174, the temperature was set at 35 ℃ for the first week, which was higher for the starter broilers in the first week. Please specify the decrease program for the temperature control in line 175.
Response
Thank you for reminding me of this comment. This change highlighted by yellow as following:
“At 1 day of age, the temperature was set at 35 °C and gradually decreased by 1 °C every 2 days until a permanent temperature of 22 °C was reached”. Then, it was maintained until the end of the trial.
Comment
Line 189, please provide the information of Sacox product, not showing this product name in other places.
Response
the information of Sacox product are provide in Lin 170 as following. “Sacox®, a commercial name for (salinomycin), is standard product protects birds from coccidiosis”.
Results
Line 298, please provide enough information including the Table to show the anti-coccidial activity of CBP.
Response
All information, including the Table demonstrating CBP's anticoccidial activity, is available in our previous research, which we refer to in the same line (Qaid et al., 2021a).
Comment
Line 303, please provide enough information of composition of cinnamon bark extract in the Material and Methods part.
Response
We provided it in Materials and Methods section 2.5. Preparation and Compositions Cinnamon Bark Powder as following “A total of 26 different active compounds with the highest quality were detected by GC-MS in the CNB extract, particularly Cinnamaldehyde, 3-phenyl-, hexadecanoic acid, (E)-2-propenal, methyl ester, 14-methyl-,methyl ester, pentadecanoic acid, oxime-, methoxy-phenyl-, and 2-methyl-benzofuran as referred previously (Qaid et al., 2021b)”.
Comment
Line 308, are provided should be changed into are shown
Response
The word “are provided” changed into “are shown”
Comment
Line 308, the sentence is not correct. There is grammar error, two verbs.
Response
Authors re wrote grammar of this sentence. Thank you.
Comment
Line 310, there is a grammar error in this sentence, did not different should be changed into did not differ or there were not significant differences in
Response
Authors replace this sentence into “Except for liver, leg, fat and lymphoid organs (bursa, thymus, and spleen) values, there were not significant differences in slaughter variables between treatments.
Comment
Line 437, there is error in this sentence. On the gastrointestinal tracts and nutrient absorption, which parameters represent these functions?? On the gastrointestinal tracts and nutrient absorption or on the most carcass traits ?? confused to the readers.
Response
Thank you so much for valuable comment. Authors changed that error in this sentence into “Here, the positive impact of cinnamon additives, particularly at level 4 g, was obvious on the most carcass traits such as the CW, carcass yield as well as breast, heart, proventriculus, gizzard, and pancreas percentage relative to CW, increased as compared to the PC”.
Comment
Line 440 delete were, changed into increased as compared to the PC
Response
Done as requested. The word “were” deleted and added “increased as compared to the PC”
Comment
Line 441, this should be changed into this result may be attributed to that cinnamon can improve the digestive system. please provide the reference.
Response
Done as requested. This result may be attributed to that cinnamon can improve the digestive system due to presence Cinnamon's active components, such as cinnamaldehyde, which can stimulate appetite and digestion (Kumar et al., 2014).
Tables,
Comment
I suggest the authors analyzed the trend of responsive indices to the dosage of cinnamon.
Response
Thank you for your advice. Of course, trend analysis using orthogonal polynomials is used when researchers want to study a trait's response to a change at different levels to a quantitative factor in order to determine the type of relationship (linear, quadratic, or cubic relationship) between the studied trait and the levels of treatment. However, in this study, authors compared cinnamon dosages to infected broiler groups treated with Sacox or to uninfected broiler groups that received a commercial diet alone (NC), or to infected broiler groups that received a commercial diet alone (PC). Thus, Duncan's multiple range test was used to determine whether there were significant differences between measurement means.
Thanks so much
Round 2
Reviewer 1 Report
All tables included only one SEM. However, each value requires each SEM for statistical analysis. Please add SEM to each value. In case of table 1, each value NC, PC, Sacox and cinnamon groups (2, 4, 6g) needs SEM. How is only one SEM to use for 6 groups ?
Author Response
Author's Reply to Reviewer 1: Round 2
Comments and Suggestions for Authors:
All tables included only one SEM. However, each value requires each SEM for statistical analysis. Please add SEM to each value. In the case of table 1, each value NC, PC, Sacox, and cinnamon groups (2, 4, 6g) needs SEM. How is only one SEM to use for 6 groups?
Response
Thank you so much for your insightful comments, which have greatly aided us in improving our manuscript. As a result, the authors have taken into account your suggestions. Therefore, each value's SEM is provided in all tables, as requested. In addition, the English language and style were fine-tuned.
Reviewer 2 Report
The author must delete the Sacox® in the abstract and other places, only showing it in the first appearance of salinomycin.
Author Response
Author's Reply to Reviewer 2: Round 2
Comments and Suggestions for Authors:
The author must delete the Sacox® in the abstract and other places, only showing it in the first appearance of salinomycin.
Response
The author deleted the Sacox® in the abstract and other places, only showing it in the first appearance of salinomycin. In addition, fine editing of English language and style revised.
Thank you for your time and effort in reviewing our manuscript and providing us with all of your comments and suggestions.